# Lysophosphatidic Acid Stimulates Mitogenic Activity and Signaling in Human Neuroblastoma Cells through a Crosstalk with Anaplastic Lymphoma Kinase

**DOI:** 10.3390/biom14060631

**Published:** 2024-05-28

**Authors:** Simona Dedoni, Maria C. Olianas, Pierluigi Onali

**Affiliations:** Laboratory of Cellular and Molecular Pharmacology, Section of Neurosciences, Department of Biomedical Sciences, University of Cagliari, Monserrato, 09042 Cagliari, Italy; dedoni@unica.it (S.D.); mariolina.olianas@gmail.com (M.C.O.)

**Keywords:** lysophosphatidic acid, anaplastic lymphoma kinase, extracellular signal-regulated kinases 1 and 2, FoxO3a, cell proliferation, human neuroblastoma cells

## Abstract

Lysophosphatidic acid (LPA) is a well-documented pro-oncogenic factor in different cancers, but relatively little is known on its biological activity in neuroblastoma. The LPA effects and the participation of the tyrosine kinase receptor anaplastic lymphoma kinase (ALK) in LPA mitogenic signaling were studied in human neuroblastoma cell lines. We used light microscopy and [^3^H]-thymidine incorporation to determine cell proliferation, Western blot to study intracellular signaling, and pharmacological and molecular tools to examine the role of ALK. We found that LPA stimulated the growth of human neuroblastoma cells, as indicated by the enhanced cell number, clonogenic activity, and DNA synthesis. These effects were curtailed by the selective ALK inhibitors NPV-TAE684 and alectinib. In a panel of human neuroblastoma cell lines harboring different ALK genomic status, the ALK inhibitors suppressed LPA-induced phosphorylation of extracellular signal-regulated kinases 1/2 (ERK1/2), which are major regulators of cell proliferation. ALK depletion by siRNA treatment attenuated LPA-induced ERK1/2 activation. LPA enhanced ALK phosphorylation and potentiated ALK activation by the ALK ligand FAM150B. LPA enhanced the inhibitory phosphorylation of the tumor suppressor FoxO3a, and this response was impaired by the ALK inhibitors. These results indicate that LPA stimulates mitogenesis of human neuroblastoma cells through a crosstalk with ALK.

## 1. Introduction

Besides being characterized as a regulator of critical physiological processes in the central nervous system and peripheral tissues, the bioactive glycerophospholipid lysophosphatidic acid (LPA) is also considered to play a role in different diseases, including cancer [1,2,3,4]. LPA activates at least six distinct G protein-coupled receptors (GPCRs), termed LPA_1–6_, which regulate a wide array of intracellular signaling pathways known to trigger mitogenesis, cell migration and survival, cytoskeleton reorganization, and angiogenesis [5]. There is evidence that in some cells, LPA-induced mitogenic signaling through extracellular signal-regulated kinases 1 and 2 (ERK1/2) involves the transactivation of receptor tyrosine kinases, such as the receptors of epidermal growth factor (EGF) and insulin-like growth factor-1 (IGF-1) [6,7,8,9].

So far, few studies have examined the effects of LPA on the biological activity of neuroblastoma, the most common extra-cranial solid tumor in children [10]. The exposure to LPA has been shown to promote proliferation of rat neuroblastoma B103 cells transfected with either *lpa*_1_, *lpa*_2_ or *lpa*_3_ receptor gene and to inhibit the motility and migration of cells expressing LPA_1_ receptor [11]. A recent study reported that LPA decreased the migration of human neuroblastoma cells but had no effect on proliferation [12]. In the latter study, however, the effects of LPA on cell proliferation were apparently investigated by incubating the cells in a medium supplemented with serum, which contains conspicuous amounts of LPA [13], making the interpretation of the results difficult. Thus, whether LPA affects the growth of human neuroblastoma cells and whether, in these cells, LPA signaling involves the activity of receptor tyrosine kinases remain to be determined.

Anaplastic lymphoma kinase (ALK) is a receptor tyrosine kinase that has been identified as a major oncogenic driver in human neuroblastoma [14,15]. ALK has been reported to be overexpressed in neuroblastoma tumors [16], and genomic rearrangements or point mutations in its catalytic domain, frequently involving F1174 and R1275, have been linked to constitutive tyrosine kinase activity and increased oncogenic potential [17,18,19,20]. Genetic dysregulation of ALK is often associated with amplification of the oncogene *MYCN* [21,22,23,24], which is detected in several cases of advanced stage tumors and considered a poor prognostic factor in neuroblastoma [10]. Based on sequence homology, ALK is classified as a member of the insulin receptor family, which also includes the leukocyte tyrosine kinase (LTK) and the IGF-1 receptor (IGF-1R). In Chinese hamster ovary fibroblasts LPA has previously been found to induce proliferation by transactivating IGF-1R [9]. However, whether a crosstalk between LPA and other members of the insulin receptor family occurs in human neuroblastoma cells has not yet been investigated.

In the present study, we report that LPA promotes the growth of human neuroblastoma cells and that the LPA mitogenic activity and signaling are curtailed by inhibition of ALK.

## 2. Materials and Methods

### 2.1. Compounds and Reagents

[Methyl-^3^H]-thymidine (20 Ci/mmol) (cat. no. NET027X) was obtained from PerkinElmer (Boston, MA, USA). 1-Oleoyl-lysophosphatidic acid (LPA) was purchased from either Santa Cruz Biotechnology (Dallas, TX, USA, cat. no. sc-222720) or Sigma-Aldrich (St. Louis, MO, USA, cat. no. L-7260). NVP-TAE684 (TAE684) (cat. no. HY-10192) and alectinib (also termed as CH5424802) (cat. no. HY-13011A) were from MedChem Express Europe (Sollentuna, Sweden). Carbachol chloride (CCh) was from Sigma-Aldrich (cat. no. C-4382). Human recombinant EGF (hEGF) (cat. no. 72528) and human recombinant IGF-1 (cat. no. 11343316) were from Cell Signaling Technology (Beverly, MA, USA) and Immuno Tools GmbH (Friesoythe, Germany), respectively. Tyrphostin AG1478 (AG1478) (cat. no. 1276/10) and PD98059 (cat no. 1213) were obtained from Tocris Bioscience (Bristol, UK). *Bordetella* pertussis toxin (PTX) (cat. no. PHZ1174) was from Gibco/Life Technologies, (Frederick, MD, USA).

### 2.2. Cell Culture

Human neuroblastoma cell lines SH-SY5Y, LAN-1, and BE(2)C were obtained from the European Collection of Authenticated Cell Cultures (ECACC) (Salisbury, UK), whereas the cell lines Kelly and NB1 were from CLS Cell Lines Service GmbH (Eppelheim, Germany) and JCRB Cell Bank (Osaka, Japan), respectively. The cell lines were authenticated by the vendors. SH-SY5Y, BE(2)C, and LAN-1 cells were grown in Ham’s F12/MEM medium (1:1) (Sigma-Aldrich) containing 2 mM L-glutamine (Sigma-Aldrich) and 1% non-essential amino acids (NEAA) (Sigma-Aldrich). Kelly and NB1 cells were cultured in RPMI 1640 containing 2 mM L-glutamine (Sigma-Aldrich). Hep2G cells from human epatocellular carcinoma (ECACC) were grown in MEM medium supplemented with 2 mM L-glutamine and 1% NEAA, whereas human embryonic kidney 293 (HEK-293) cells (ECACC) were cultured in Ham’s F12/DMEM medium (1:1) containing 2 mM L-glutamine. Media were supplemented with 10% foetal calf serum (FCS) and 100 U/mL penicillin-100 µg/mL streptomycin (Sigma-Aldrich). Cells were maintained at 37 °C in a humidified atmosphere of 5% CO_2_ in air. Sub-confluent cultures were split every 72 h and seeded at the density of 1–3 × 10^4^/cm^2^ using 0.25% trypsin/EDTA (Sigma-Aldrich). After resuscitation, cells were used for no more than 10–15 passages. Mycoplasma contamination was periodically checked by using the MycoFluor Mycoplasma Detection kit (Invitrogen-Life Technologies, Carlsbad, CA, USA). None of the cell lines resulted positive.

### 2.3. Cell Treatment and Cell Lysate Preparation

Unless otherwise specified, cells were washed with phosphate-buffered saline (PBS) (pH 7.4) and incubated overnight in serum-free medium. Cells were treated with the test agents as indicated in the text and maintained at 37 °C in a humidified atmosphere of 5% CO_2_ in air. The time periods of cell exposure to either TAE684 or alectinib and the concentrations of these agents were chosen on the basis of the results of previous studies [25,26]. For PTX treatment, cells were incubated overnight in serum-free medium containing either vehicle or 100 ng/mL of the toxin. Thereafter, the medium was renewed, and the cells were treated with the test agents as indicated. Cell lysates were prepared by washing with PBS and scraping the cells into an ice-cold lysis buffer containing PBS, 0.1% sodium dodecyl sulphate (SDS), 1% Nonidet P-40, 0.5% sodium deoxycholate, 2 mM EDTA, 2 mM EGTA, 4 mM sodium pyrophosphate, 2 mM sodium orthovanadate, 10 mM sodium fluoride, 20 nM okadaic acid, 1 mM phenylmethylsulphonyl fluoride (PMSF), 0.5% phosphatase inhibitor cocktail 3, and 1% protease inhibitor cocktail (Sigma-Aldrich) (RIPA buffer). The samples were sonicated for 5 s in ice-bath, and aliquots of cell extracts were taken for protein determination by the Bio-Rad protein assay (Bio-Rad Lab, Hercules, CA, USA).

### 2.4. Cell Transfection with Small Interfering RNA (siRNA)

SH-SY5Y cells were grown in 6-well plates to 50–60% confluency. Cells were washed with PBS and incubated with Opti-MEM I reduced serum medium (Invitrogen-Life Technologies). Cells were transfected with either ALK siRNA (50 nM) (cat. no. SR300165) or control siRNA (50 nM) (cat. no. SR30004) (Origene, Rockville, MD, USA) using Lipofectamine RNAi MAX (Invitrogen-Life Technologies) as a transfectant. The ALK siRNA comprised three unique 27mer siRNA duplexes (Appendix A). After 5 h, the medium was removed, and the cells were incubated in growth medium for 48 h. Thereafter, the cells were incubated overnight in serum-free medium and used for the experiments.

### 2.5. Generation of Cells Individually Expressing LPA_1_, LPA_2_, and LPA_3_ Receptors

Human cDNAs cloned into pcDNA3.1 vector obtained from UMR cDNA Resource Center (Rolla, MO, USA) and cDNA Resource Center (Bloomsburg, PA, USA; www.cdna.org, accessed on 30 January 2018) were used to generate HEK-293 cells overexpressing human LPA_1_ (HEK-LPA_1_), LPA_2_ (HEK-LPA_2_), and LPA_3_ (HEK-LPA_3_) receptors. Lipofectamine 2000 (Invitrogen-Life Technologies) was employed as a transfectant. Stably transfected cell clones were selected by incubation for six to seven weeks with 1 mg/mL G418 sulfate and subsequent screening for LPA-induced phospho-ERK1/2 stimulation. Positive clones were maintained in growth medium supplemented with 0.4 mg/mL G 418 sulfate.

### 2.6. Generation of FAM150B Containing Conditioned Medium

FAM150B (also designated Augmentor-α or ALKAL2) containing conditioned medium was generated as previously described [27] with some modifications. HEK-293 were seeded in 100 mm plastic Petri dishes and grown to ~70% confluency. Cells were transfected in Opti MEM I medium with 2 µg/mL of pEZ-M02 vector containing the open reading frame of human FAM150B (cat. no. H0627) (GeneCopoeia, Rockville, MD, USA) using Lipofectamine 2000. After 7 h, the cells were washed and incubated in growth medium supplemented with G 418 sulfate to select stably transfected clones. Cell colonies were isolated and grown to near confluency. Cells were incubated for 24 h in serum-free medium to obtain the conditioned medium. The medium of each clone was centrifuged at 500× *g* for 10 min, and the supernatant was passed through a sterile 0.45 µm filter (Millipore, Bedford, MA, USA) and examined for the ability to induce ALK phosphorylation.

### 2.7. Immunoprecipitation

Serum-starved SH-SY5Y cells were treated for 5 min with either vehicle or LPA, lysed with ice-cold RIPA buffer supplemented with 1% Triton × 100 and incubated for 30 min at ice-bath temperature. Following centrifugation at 10,000× *g* for 10 min at 4 °C, the supernatant (~500 µg of protein) was incubated overnight at 4 °C with either anti-ALK antibody (cat. no. 3333, Cell Signaling Technology, RRID: AB_836862) (1:100) or pre-immune rabbit IgG (1:100) (cat. no.12-370 Sigma-Aldrich). Thereafter, 50 µL of Pure Proteome Protein G magnetic beads (Millipore, Burlington, MA, USA) were added, and samples were incubated at 4 °C for 3 h with continuous rotation at 4 °C. The beads were washed 5 times with ice-cold PBS/0.1% Tween 20 buffer. After the last wash, the beads were resuspended in 2× sample buffer and boiled for 5 min.

### 2.8. Western Blot Analysis

Aliquots of cell lysates containing an equal amount of protein were subjected to SDS-polyacrylamide gel electrophoresis and, then, electrophoretically transferred to polyvinylidene difluoride membranes. Membranes were blocked with 5% non-fat dry milk (Santa Cruz Biotechnology), washed and incubated overnight at 4 °C with one of the following primary antibodies: rabbit polyclonal anti-phospho-ERK 1(Thr202/Tyr204)/ERK2 (Thr185/Tyr187) (cat. no. RA15002, Neuromics, Northfield, MN, USA, RRID: AB_2139954) (1:10,000), rabbit polyclonal anti-ERK1/2 (cat. no. 9102, Cell Signaling Technology, RRID: AB_330744) (1:1000), rabbit monoclonal anti-ALK (cat. no. 3333, Cell Signaling Technology, RRID: AB_836862) (1:1000), rabbit polyclonal anti-phospho-ALK (Tyr1604) (cat. no. 3341, Cell Signaling Technology, RRID: AB_331047) (1:1000), rabbit monoclonal anti-phospho-FoxO3a (Ser425) (cat. no. 64616, Cell Signaling Technology, RRID: AB_2799662) (1:1000), rabbit polyclonal anti-FoxO3a (cat no, 10849-1-AP, Proteintech, Rosemont, IL, USA, RRID: AB_2247214) (1:2000), mouse monoclonal anti-phospho-Tyrosine (cat. no. 9411, Cell Signaling Technology, RRID: AB_331228) (1: 2000). As a loading control, glyceraldehyde 3-phosphate dehydrogenase (GAPDH), detected with a rabbit polyclonal antibody (cat. no. 247002, Synaptic Systems, Gottingen, Germany, RRID: AB_10804053) (1:10,000), or actin, detected with a rabbit polyclonal antibody (cat. no. A2066, Sigma-Aldrich, RRID: AB_476693) (1:3000), was used. Thereafter, the membranes were washed and incubated with horseradish peroxidase-conjugated goat anti-rabbit IgG (A 0545 Sigma-Aldrich) (1:15,000) for 1 h at room temperature. Immunoreactive bands were detected by using Clarity Western ECL substrate (Bio-Rad Lab), and digital images were obtained by using either ECL Hyperfilm (Amersham, Piscataway, NJ, USA) with Image Scanner III (GE Healthcare, Milan, Italy) or chemiluminescence Image analyzer LAS 4000 (FujiFilm, Tokyo, Japan). Band densities were determined by using the NIH ImageJ software, v1.54b (US National Institutes of Health, Bethesda, MD, USA). The density of the phosphorylated protein bands was normalized to the density of the corresponding total protein in the same sample.

### 2.9. Cell Counting

SH-SY5Y cells were seeded into 12-well plates at a density of 20 × 10^3^ cells/well and grown to ~30% confluency. The growth medium was removed, and the cells were incubated for 18 h in a serum-free medium. Thereafter, the serum-free medium was renewed, the test agents were added, and the incubation continued for 24 h. Cells were washed with PBS, fixed in 4% paraformaldehyde, and digital images were taken in five fields/well by phase contrast microscopy using an Olympus IX 51 (Olympus Europe GmbH, Hamburg, Germany) inverted microscope equipped with a Plan achromatic 20× objective. Cells present in each field were counted using the software Cell P, 1.0 (Olympus Soft imaging Solutions) by an investigator unaware of the treatment.

### 2.10. Clonogenic Assay

SH-SY5Y cells were detached by treatment with trypsin-EDTA solution, and single cell suspensions were seeded into 12-well plates at a density of 100 cells/well in growth medium. After attachment, cells were washed, incubated in Ham’s F12/MEM medium containing 2 mM L-glutamine, 1% NEAA, and 10% active charcoal-treated FCS (Sigma-Aldrich) and treated as specified in the text. The medium and the test agents were renewed every other day. After one week, colonies were washed with PBS, fixed with 10% neutral buffered formalin, and stained with 0.01% crystal violet. Digital images of the colonies were obtained using a camera device, and colonies were counted using the ImageJ software, v1.54b.

### 2.11. [^3^H]-Thymidine Incorporation

Cells were grown to about 50% confluency in 24 well plates, washed and incubated in serum-free medium for 12–18 h. Thereafter, the serum-free medium was replaced, and the cells were treated with the test compounds as indicated in the text. One hour after LPA addition, cells were incubated for 24 h with [^3^H]-thymidine (0.25 µCi/well). Following incubation, the medium was removed, and the cells were washed twice with 5% ice-cold trichloroacetic acid. Cells were solubilized by overnight incubation with 0.1 N NaOH containing 0.5% SDS. The samples were neutralized with HCl, and [^3^H]-thymidine incorporation was determined by liquid scintillation counting. Assays were performed in triplicate.

### 2.12. Statistical Analysis

Results are reported as means ± S.D. of the indicated number of independent experiments. Statistical analysis was performed by using the program Graph Pad Prism 5 (San Diego, CA, USA), which was also used to calculate EC_50_ and E_max_ values. Unless otherwise indicated, data are expressed as percentage or fold stimulation of control, which was included in each independent experiment. The variance of the control group was obtained by expressing each control value as a percentage of the mean of the raw values of the control group. Statistical significance between experimental groups was assessed by either analysis of variance (ANOVA) followed by Neuman Keuls test or Student’s *t* test, as appropriate. A value of *p* < 0.05 was considered as the level of statistical significance.

## 3. Results

### 3.1. LPA Stimulates the Proliferation of Human Neuroblastoma Cells: Blockade by ALK Inhibitors

Light microscopy observation of cultured human SH-SY5Y neuroblastoma cells revealed that prolonged exposure (48 h) to LPA (10 µM) increased the proliferative rate, as indicated by the enhanced cell confluency when compared to vehicle-treated samples (Figure 1a). Cell pretreatment with either TAE684 (50 nM), a prototype ALK inhibitor [25], or alectinib (1 µM), a second generation ALK inhibitor [26], reduced cell growth and inhibited the stimulatory effect of LPA (Figure 1a). Quantitative analysis indicated that LPA treatment increased the cell number by 36 ± 11% (*p* < 0.01) (Figure 1a, scatter plot). Cell treatment with TAE684 and alectinib reduced the cell number by 28 ± 6% (*p* < 0.05) and 30 ± 8% (*p* < 0.05), respectively, and abrogated the enhancement elicited by LPA (Figure 1a, scatter plot).

When seeded at low density, single SH-SY5Y cells proliferate to form colonies, a behavior indicative of viability, stemness, and tumorigenicity [28,29]. As shown in Figure 1b, the addition of 1 µM LPA increased the number of colonies by 52 ± 21% (*p* < 0.001) as compared to control samples. Preliminary experiments indicated that this concentration of LPA produced a maximal stimulatory response. Cell treatment with either TAE684 (50 nM) or alectinib (100 nM) had no significant effect on colony formation but almost completely suppressed the stimulatory effect of LPA (Figure 1b, scatter plot). In these experiments, a nanomolar concentration of alectinib was employed in order to avoid cell death occurring following long-term exposure to micromolar concentrations of the drug [30].

To further investigate the mitogenic activity of LPA and its blockade by ALK inhibitors, we examined [^3^H]-thymidine incorporation, an index of DNA synthesis. In SH-SY5Y cells, LPA induced a concentration-dependent increase in [^3^H]-thymidine incorporation with an EC_50_ value of 1.6 ± 0.3 µM and an E_max_ value corresponding to 65 ± 5% increase in basal level (*p* < 0.001) (Figure 1c). Cell exposure to either TAE684 (50 nM) or alectinib (1 µM) reduced basal [^3^H]-thymidine incorporation by 22 ± 5% (*p* < 0.05) and 36 ± 3% (*p* < 0.01), respectively, and completely prevented the stimulatory effect of LPA (Figure 1d). Conversely, cell pretreatment with TAE684 (50 nM) failed to significantly affect the stimulation of [^3^H]-thymidine incorporation induced by either the cholinergic receptor agonist carbachol (CCh) (10 µM) or IGF-1 (20 ng/mL) (Appendix A). The pre-exposure to alectinib (1 µM) reduced both CCh and IGF-1 stimulations expressed as a percent of control (Appendix A). However, when calculating the net stimulations, alectinib had no significant effect on the response to CCh and caused a modest and not significant reduction in that to IGF-1. Net stimulations, expressed as dpm/well ± SD, were as follows: vehicle + CCh 5913 ± 1960, alectinib + CCh 5638 ± 1855 (*p* > 0.05); vehicle + IGF-1 10,338 ± 1822, alectinib + IGF-1 7763 ± 1053 (*p* > 0.05).

As in different cell types, LPA has been reported to promote cell proliferation through the activation of ERK1/2 [9,31,32]; we investigated whether this signaling pathway mediated the mitogenic activity of LPA in human neuroblastoma cells. As shown in Figure 1e, pretreatment of SH-SY5Y cells with PD98059 (25 µM), a selective inhibitor of ERK1/2 upstream activators MEK1/2, abrogated the stimulation of [^3^H]-thymidine incorporation induced by LPA.

### 3.2. LPA Activates ERK1/2 in Different Human Neuroblastoma Cell Lines

We next investigated whether the exposure to LPA was able to affect ERK1/2 activity in a panel of human neuroblastoma cell lines with different ALK genomic status. The panel included SH-SY5Y, Kelly, and LAN-1 cells, which harbor the gain-of function mutation F1174L, BE(2)C expressing the wild-type form of ALK [15,17,18,33], and NB1 cells bearing the amplification of an abnormal ALK gene with a deletion of exons 2 and 3 [34]. Moreover, the cell lines varied with regard to *MYCN* expression, being without (SH-SY5Y) and with (Kelly, LAN-1, BE(2)C, and NB1) gene amplification [35,36,37]. Time-course experiments showed that in each neuroblastoma cell line examined, exposure to LPA (10 µM) induced a rapid elevation in the levels of dually phosphorylated ERK1/2, indicating increased kinase activities (Figure 2). Following the initial increase, the levels of phospho-ERK1/2 rapidly declined, while remaining significantly higher than basal values up to or at 60 min of LPA exposure in SH-SY5Y, Kelly, LAN-1, and NB1 cells.

### 3.3. ALK Inhibitors Curtail LPA-Induced ERK1/2 Phosphorylation

Once observed that LPA was capable of effectively stimulating ERK1/2 phosphorylation in different human neuroblastoma cell lines, we examined whether this response was affected by ALK inhibitors. As shown in Figure 3a–e, pretreatment with TAE684 (50 nM) markedly inhibited ERK1/2 phosphorylation triggered by 1 and 10 µM LPA in SH-SY5Y, Kelly, and LAN-1 cells and completely suppressed the stimulation by 10 µM LPA in NB1 and BE(2)C cells. With the exception of Kelly cells, the exposure to TAE684 also significantly decreased the levels of phospho-ERK1/2 in vehicle-treated cells.

Similar results were obtained when the effects of alectinib were studied. Following preexposure to alectinib (1 µM), LPA-induced stimulation of ERK1/2 phosphorylation was reduced by 55 ± 6% (*p* < 0.001) and 48 ± 5% (*p* < 0.001) in SH-SY5Y and LAN-1 cells, respectively, whereas a complete LPA blockade was detected in Kelly, NB1, and BE(2)C cells (Figure 3f–j). As observed with the exposure to TAE684, cell pretreatment with alectinib significantly reduced phospho-ERK1/2 levels in vehicle-treated cells. These results are in agreement with previous studies indicating that in human neuroblastoma cells, ALK activity controls the basal state of ERK1/2 phosphorylation [38,39].

We also investigated the effect of ALK inhibition on ERK1/2 activation induced by CCh and compared this effect with that of LPA. To facilitate the comparison, cells were treated in the same experiment with either CCh or LPA. As shown in Appendix A, pretreatment of SH-SY5Y cells with TAE684 (50 nM) had no effect on ERK1/2 phosphorylation stimulated by CCh (10 µM), while it significantly inhibited the response to LPA (10 µM).

We then explored whether inhibition of receptor tyrosine kinases other than ALK affected LPA-induced ERK1/2 activation. We considered the EGF receptor as a possible target because this receptor tyrosine kinase has been shown to be transactivated by LPA in different cell systems. Pretreatment of SH-SY5Y cells with AG 1478 (200 nM), a selective inhibitor of EGF receptor tyrosine kinase activity, completely suppressed the stimulation of ERK1/2 phosphorylation induced by EGF (50 ng/mL) and had a modest inhibitory effect on the stimulation induced by 10 µM LPA (21 ± 3% decrease, *p* < 0.05) (Appendix A).

### 3.4. Effects of ALK Inhibitors on LPA-Induced ERK1/2 Activation in HEK-293 Cells Transfected with LPA_1–3_ Receptors and in HepG2 Cells

To rule out the possibility that the inhibition of LPA activity by either TAE684 or alectinib was due to off target actions at LPA receptors, we examined the effects of the two inhibitors on the stimulation of ERK1/2 phosphorylation in HEK-293 cells individually expressing LPA_1_, LPA_2_, and LPA_3_ receptors. Previous studies have shown that HEK-293 cells do not express ALK [40,41]. As shown in Figure 4a–c, the exposure to either TAE684 (50 nM) or alectinib (1 µM) induced potentiation, rather than inhibition, of ERK1/2 phosphorylation elicited by LPA at each receptor subtype. These results indicated that the suppression of LPA signaling elicited by the ALK inhibitors in human neuroblastoma cells was not due to an impairment of LPA ability to induce receptor activation.

Both TAE684 and alectinib have been shown to inhibit the activity of LTK [42,43], which is expressed in human neuroblastoma cells (The Human Protein Atlas, www.proteinatlas.org, accessed on 27 February 2024). To investigate whether LTK was involved in the inhibitory action of ALK inhibitors on LPA signaling, we used HepG2 cells, which express LTK but not ALK [43]. We found that LPA (10 µM) elicited a marked stimulation of ERK1/2 phosphorylation in these cells. Pretreatment with TAE687 (50 nM) failed to affect LPA-induced stimulation of ERK1/2 phosphorylation, whereas a potentiation of the LPA effect was observed in cells pretreated with alectinib (1 µM) (Figure 4d). These results suggest that the blockade of LTK was not associated with a reduced response to LPA.

### 3.5. Downregulation of ALK Attenuates LPA-Induced ERK1/2 Stimulation

To further assess the involvement of ALK in LPA signaling, we examined the response to the lysophospholipid in SH-SY5Y cells treated with ALK siRNA to induce a decrease in the levels of the protein. In agreement with previous studies [40,41], Western blot analysis using a primary antibody recognizing an internal sequence of the ALK protein detected two immunoreactive bands, one displaying an apparent molecular mass of approximately 220 kDa, which corresponds to the mature glycosylated form of ALK, and the other migrating at approximately 140 kDa (Figure 5a). Cell treatment with ALK siRNA caused a significant reduction in the steady-state levels of the 220 and 140 kDa molecular forms of ALK by 29 ± 5% (*p* < 0.05) and 37 ± 8% (*p* < 0.05), respectively, as compared with the levels of control siRNA-treated cells. Cells treated with either control siRNA or ALK siRNA were incubated with either LPA (10 µM) or CCh (10 µM). As shown in Figure 5b,c, ALK depletion was associated with a significant reduction in the stimulation of ERK1/2 phosphorylation induced by LPA (35 ± 10% decrease as compared to control siRNA-treated cells, *p* < 0.05) and no change in the response to CCh.

### 3.6. LPA Potentiates ALK Activation in Human Neuroblastoma Cells

We next examined whether the exposure to LPA affected endogenous ALK activity. To activate ALK, we used the conditioned medium of HEK-293 cells stably transfected with the cDNA encoding human FAM150B, which preferentially activates ALK over LTK [44]. The level of ALK phosphorylation at Tyr1604, located in the intracellular domain of the receptor, was used as an index of ALK activation [27]. As shown in Figure 6a, time- course experiments indicated that the incubation of SH-SY5Y cells with FAM150B containing conditioned medium induced a rapid and marked increase in the phosphorylation of the 220 and 140 kDa molecular forms of ALK, which lasted up to 30 and 60 min, respectively. In contrast, cell exposure to medium from non-transfected HEK-293 cells, used as control medium, failed to induce ALK phosphorylation at each time point examined (Figure 6b). The incubation of SH-SY5Y cells with FAM150B containing conditioned medium elicited a robust increase in ERK1/2 phosphorylation, which displayed a kinetic profile consistent with that shown by ALK activation (Figure 6c), and gradually decreased upon serial dilution of the conditioned medium (Figure 6d). Moreover, pretreatment with either TAE684 (50 nM) or alectinib (1 µM) abrogated the stimulation of ALK phosphorylation (Figure 6e) and ERK1/2 activation in SH-SY5Y cells (Figure 6g). Consistent with a previous study [34], a Western blot analysis of NB1 cell lysates showed that the higher molecular form of ALK had an apparent molecular mass of 208 kDa due to gene deletions (Figure 6f). Both molecular forms of ALK displayed a high level of phosphorylation under basal conditions, which was further increased by incubation with FAM150B containing conditioned medium. Pretreatment with the ALK inhibitors suppressed both basal and stimulated ALK phosphorylations (Figure 6f).

We then investigated the effect of LPA on ALK activation by FAM150B containing conditioned medium. Pretreatment of SH-SY5Y cells with LPA (10 µM) potentiated the phosphorylation of the 140 kDa form of ALK induced by FAM150B containing conditioned medium, while causing a small and nonsignificant increase in the phosphorylation of the 220 kDa form (Figure 6h). In the same experiments, LPA was found to enhance the stimulation of ERK1/2 phosphorylation elicited by FAM150B containing conditioned medium (Figure 6i). Cell pretreatment with PTX, which uncouples receptors from G proteins of the Gi/o family, prevented the activation of ERK1/2 by LPA and the LPA-induced potentiation of ERK1/2 and ALK phosphorylations elicited by FAM150B containing medium (Appendix A).

We explored whether LPA affected the phosphorylation state of ALK in the absence of ALK stimulation. To this goal, SH-SY5Y cells were treated with either vehicle or LPA (10 µM), ALK was immunoprecipitated from cell extracts, and immunoprecipitates were analyzed by a Western blot with an antibody directed against phospho-Tyrosine residues. As shown in Figure 6j, a brief exposure to LPA increased the tyrosine phosphorylation of the 140 kDa form of ALK while having a nonsignificant effect on the phosphorylation state of the 220 kDa form.

### 3.7. LPA Regulates FoxO3a Phosphorylation and Expression through ERK1/2

FoxO3a is a member of the Forkhead family of transcription factors which act as tumor suppressors by inducing cell cycle arrest and apoptotic cell death [45]. It has been demonstrated that ERK1/2 phosphorylate FoxO3a at specific serine residues, including Ser425, thereby promoting FoxO3a degradation [46]. To investigate the participation of FoxO3a in the mitogenic activity of LPA in human neuroblastoma cells, we first examined the ability of LPA to regulate the phosphorylation state of the transcription factor. Time-course experiments showed that in SH-SY5Y cells, LPA (10 µM) induced an elevation of FoxO3a phosphorylated at Ser425 which displayed a lag of 15 min and reached a maximum at about 60 min (Figure 7a). Prolonged exposure (24 h) to LPA (10 µM) caused a significant decrease in the cellular content of FoxO3a protein (60 ± 20% reduction in control levels, *p* < 0.05) (Figure 7b). Pretreatment of SH-SY5Y cells with PD98059 (25 µM) reduced basal levels of phosphorylated FoxO3a and completely blocked the stimulatory effect of LPA (Figure 7c). When the effect of ALK inhibition was examined, we found that exposure of SH-SY5Y cells to either TAE684 (50 nM) or alectinib (1 µM) reduced the increase in LPA-induced phospho-FoxO3a levels by 59 ± 16 (*p* < 0.001) and 46 ± 14% (*p* < 0.001), respectively (Figure 7d).

## 4. Discussion

A first important finding of the present study is the observation that LPA promotes the growth of cultured human neuroblastoma cells. In SH-SY5Y cells, an established cell line originally derived from a bone marrow biopsy, the exposure to LPA produced a significant increase in cell number, clonogenicity, and DNA synthesis. Previous studies examining the expression of LPA receptor subtypes in SH-SY5Y cells by quantitative RT-PCR have demonstrated the predominant presence of LPA_2_ receptor mRNA, a lower level of LPA_3_, and the absence of LPA_1_ receptor transcripts [47]. The magnitude of proliferative responses to LPA was generally higher than that previously observed in rat neuroblastoma B103 transfected with LPA_1–3_ receptors [11], suggesting that LPA receptors expressed under native conditions stimulate cell growth more effectively than individual receptor subtypes ectopically expressed in host cells. However, the cell growth elicited by LPA in SH-SY5Y cells appeared much smaller than that induced by the lysophospholipid via endogenous LPA_2_ and LPA_3_ receptors in HCT116 and LS174T colon cancer cells [48], indicating that, besides the pattern of LPA receptor subtypes expressed, additional biological properties of each cancer cell type are relevant in determining the strength of the proliferative response. However, it is important to note that the concentrations of LPA inducing mitogenesis of SH-SY5Y cells (1–10 µM) were in the same range of those required to stimulate the proliferation of rat neuroblastoma [11] and ovarian and breast cancer cells [49,50]. The mitogenic activity of LPA appeared to involve the activity of the ERK1/2 signaling pathway, and the lysophospholipid behaved as an effective stimulator of ERK1/2 phosphorylation in all human neuroblastoma cell lines examined. Moreover, in most of the cells, the stimulation of phospho-ERK1/2 levels appeared to be sustained, a condition that has been found to be required for G1-to-S-phase progression of the cell cycle [51]. The involvement of ERK activation in the proliferative response to LPA has also been documented in other cancers, such as breast cancer and endometrial carcinoma cells [52,53].

Previous studies in human neuroblastoma have found that FoxO3a mRNA levels are lower in high-stage than in low-stage tumors and that a low expression of FoxO3a is associated with a poor prognosis [54,55]. FoxO3a is known to enhance the activity of genes promoting cell cycle arrest (*p21^Cip1^*, *p27^Kip1^*, and *p57^Kip2^*) and apoptosis (*Bim*, *Puma*, and *Noxa*) [45]. Post-translational modifications are key events in the regulation of FoxO protein transcriptional activity [56]. Among these changes, phosphorylation by signaling kinases, such as ERK1/2, protein kinase B/Akt, serum- and glucocorticoid-inducible kinase, and IkB kinase β, has been demonstrated to control the cellular fate of FoxO3a [56]. Upon phosphorylation, nuclear FoxO3a binds to 14-3-3 protein with the consequent translocation to the cytoplasm where it is degraded [45]. In particular, ERK1/2 interacts directly with FoxO3a, and the consequent phosphorylation of FoxO3a promotes its degradation by the MDM2-mediated ubiquitin-proteasome pathway [46]. In the present study, we found that in SH-SY5Y cells, LPA stimulated the phosphorylation of FoxO3a in a manner sensitive to blockade by the MEK1/2 inhibitor PD98059. LPA-induced phospho-FoxO3a accumulation was delayed as compared to the rapid ERK1/2 stimulation, likely reflecting the time interval required by activated ERK1/2 to reach the nucleus and was associated with a significant drop in the cellular levels of FoxO3a following prolonged exposure to LPA. Collectively, these results identify FoxO3a as a relevant downstream target of ERK1/2 activation by LPA in human neuroblastoma cells.

An additional novel finding of the present study is the demonstration that ALK is a major player in mediating the mitogenic signaling of LPA in human neuroblastoma cells. Pharmacological evidence indicates that the downregulation of ALK activity curtailed LPA-induced proliferation and ERK1/2 activation. TAE684 has been shown to display high selectivity for ALK over a panel of other tyrosine kinases and to affect insulin-induced cellular signaling only at concentrations about 100-fold higher than those inhibiting ALK [25]. Alectinib has been developed as a next generation highly selective inhibitor of several mutated forms of ALK [26] and has been approved for the treatment of ALK rearranged lung cancers [57]. In SH-SY5Y cells, both TAE684 and alectinib completely suppressed the increase in cell number, clonogenicity, and DNA synthesis induced by LPA. Under similar experimental conditions, the ALK inhibitors had nonsignificant effects on the mitogenic responses to CCh and IGF-1, showing some selectivity for LPA activity. Both TAE684 and alectinib were found to be effective inhibitors of LPA-induced ERK1/2 activation in human neuroblastoma cell lines known to bear different ALK and NMYC genomic status. Conversely, complete blockade of EGF receptor activity was found to cause a minor inhibition of LPA-induced ERK1/2 activation in SH-SY5Y cells, indicating that, at least in these cells, this tyrosine kinase receptor was not the predominant LPA partner.

The suppression of ERK1/2 activation by the ALK inhibitors resulted in a dampening of FoxO3a phosphorylation with potential impact on the transcriptional mechanisms regulating mitogenesis and apoptosis. Different experimental outcomes supported the idea that TAE684 and alectinib counteracted LPA-induced ERK1/2 activation by specifically acting as ALK inhibitors. First, in SH-SY5Y cells, their inhibitory action on LPA signaling was reproduced by down regulating ALK expression with siRNA-treatment. In the same experiment, ALK depletion did not affect the stimulation of ERK1/2 by CCh, in line with the observation that ALK activity was dispensable for the mitogenic response to the muscarinic receptor agonist. The inhibition of ALK expression obtained following siRNA treatment was only partial, accounting for about 30–35% decrease in control protein levels. This represents a limitation of the present study, and additional work in cells displaying a greater level of ALK depletion is necessary to further substantiate this point. However, it should be noted that in human neuroblastoma cells, a near-complete or complete knockdown of ALK has been found to be associated with apoptotic death, particularly in SH-SY5Y cells [18,58]. Second, the ALK inhibitors were found to have no effect or to cause a potentiation, rather than inhibition, of LPA-induced ERK1/2 activation in cell lines lacking ALK expression, such as HEK-293 and Hep2G cells, indicating that their inhibitory effect on LPA signaling required the presence of ALK. At present, the precise reasons why, in these cells, the response to LPA was enhanced by exposure to the ALK inhibitors are not known.

The occurrence of a functional interaction between LPA and ALK in human neuroblastoma cells was further underlined by the finding that LPA potentiated ALK activation by FAM150B conditioned medium, as indicated by the enhanced receptor phosphorylation and the synergistic interaction in inducing ERK1/2 activation. In both NB1 and SH-SY5Y cells, ALK activation by FAM150B conditioned medium was abrogated by pretreatment with either TAE684 or alectinib, demonstrating the specificity of the observed effects. The intracellular pathways linking LPA receptor activation to ALK phosphorylation remain to be investigated. The results obtained with PTX indicate that LPA-induced ERK1/2 activation involved the activity of G proteins of the Gi/Go family.

PTX-sensitive G proteins have been reported to transduce LPA mitogenic signaling also in other cell types [9,31,59]. The βγ subunits of PTX-sensitive G proteins have been shown to activate the non-receptor tyrosine kinase c-Src [60], which can act as an intracellular mediator of ligand-independent transactivation of tyrosine kinase receptors triggered by GPCR [8]. Notably, Src can phosphorylate IGF-1R at sites that are the same as ligand-induced autophosphorylation [61], and we have previously reported that Src mediates LPA-induced tyrosine phosphorylation and activation of IGF-1R in CHO-K1 fibroblasts [9]. Given the high structural homology between IGF-1R and ALK, a possible intracellular mechanism by which LPA phosphorylates ALK may involve the activation of Src.

LPA was found to preferentially enhance the phosphorylation of the 140 kDa form of ALK over that of the 220 KDa form, suggesting that the former molecular species is the one mostly involved in the mitogenic signaling of the lysophospholipid. Although the relative biological role of the two molecular forms of ALK is not fully known, both appear to be activated by cell exposure to FAM150B containing conditioned medium, indicating a common contribution to ALK signaling. It has been reported that the 140 kDa form originates from the extracellular proteolytic cleavage of the N-terminal portion of the 220 kDa form [40]. The extracellular domain shedding of the 220 KDa form has been demonstrated to be mediated by matrix metalloprotease 9 (MMP-9) and to be associated with enhanced neuroblastoma cell migration [41]. In this regard, it is noteworthy that LPA is known to induce the expression and activation of different types of matrix metalloproteases, including MMP-9 [62,63,64]. Thus, an intriguing possibility, which remains to be investigated, is that LPA not only favors ALK activation but may also promote the extracellular domain shedding of the receptor.

## 5. Conclusions

In conclusion, the present study shows that LPA enhances the proliferation of human neuroblastoma cells by activating endogenous receptors which trigger mitogenic signals through crosstalk with ALK. Previous studies comparing gene expression in various tumors versus normal tissues have revealed a high level of LPA receptors, particularly LPA_1_ and LPA_2_, in human neuroblastoma specimens [64]. Moreover, the expression of autotaxin, the major LPA-generating enzyme, has also been found to be elevated in neuroblastoma cells [64,65], indicating that LPA may be a critical component of the tumor microenvironment. Taken together, the present study suggests that pharmacological interventions aimed to disrupt the LPA-ALK interplay may provide a beneficial outcome in constraining neuroblastoma growth.

## Figures and Tables

**Figure 1 biomolecules-14-00631-f001:**
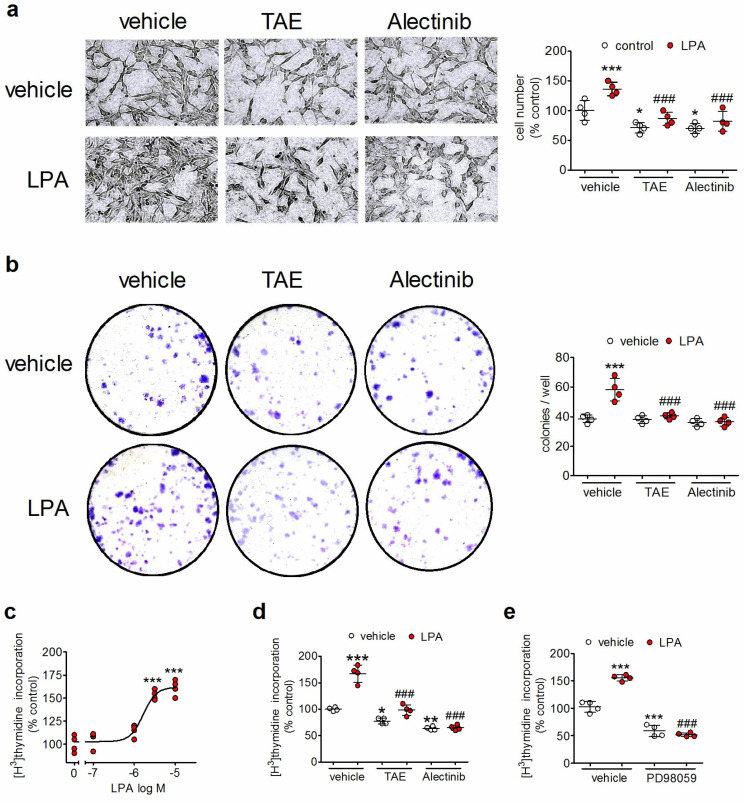
ALK inhibitors impair LPA-induced mitogenesis of human neuroblastoma cells. (**a**) Microscopic analysis of serum-starved SH-SY5Y cells pretreated with either vehicle, 50 nM TAE684 (TAE) for 1 h, or 1 µM alectinib for 6 h, and then incubated for additional 24 h with either vehicle or 10 µM LPA. The scatter plot reports the quantitation of cell number in each experimental group. (**b**) Colony formation by SH-SY5Y cells incubated for one week in the presence of either vehicle, 50 nM TAE684, 100 nM alectinib, 1 µM LPA, or the combination of LPA with the ALK inhibitors. (**c**) Serum-starved SH-SY5Y were treated with either vehicle or the indicated concentrations of LPA. [^3^H]-Thymidine was then added, and the incubation was continued for 24 h. (**d**) Serum-starved cells were treated as indicated in (**a**) and incubated in the presence of [^3^H]-thymidine for 24 h. (**e**) Serum-starved SH-SY5Y cells were pretreated for 1 h with either vehicle or 25 µM PD98059, and then incubated for 24 h with either vehicle or 10 µM LPA in the presence of [^3^H]-thymidine. Values are the mean ± SD of four independent experiments. * *p* < 0.05, ** *p* < 0.01, *** *p* < 0.001 vs. control (vehicle-treated cells). ^###^ *p* < 0.001 vs. vehicle + LPA by ANOVA followed by Neuman Keuls test.

**Figure 2 biomolecules-14-00631-f002:**
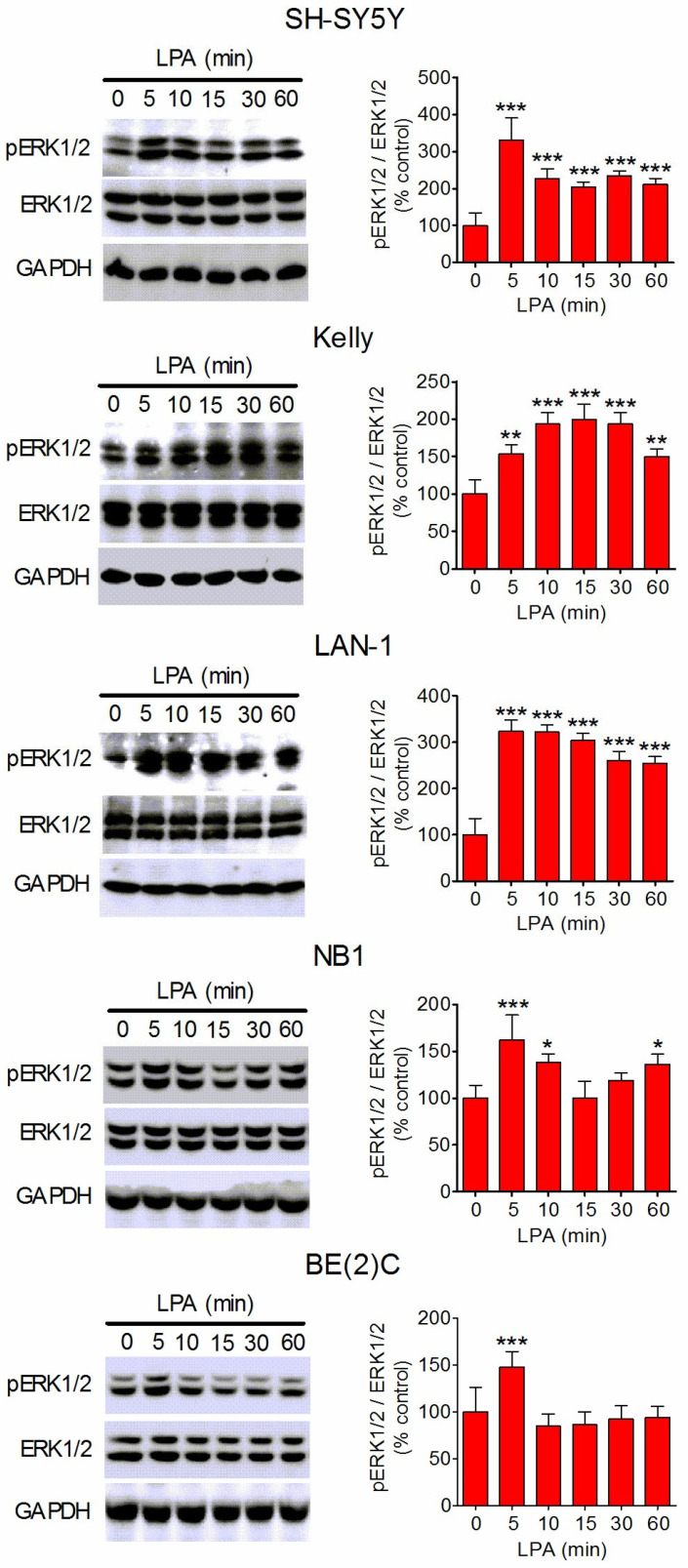
Time-dependent stimulation of ERK1/2 phosphorylation by LPA in different human neuroblastoma cell lines. Serum-starved cells were incubated for the indicated periods of time with LPA (1 µM). Zero time samples were treated with vehicle and used as control. Thereafter, cell lysates were analyzed for phospho-ERK1/2 (pERK1/2), total ERK1/2, and GAPDH by Western blot. Densitometric ratios of pERK1/2/ERK1/2 are reported as percent of control. Values are the mean ± SD of three independent experiments. * *p* < 0.05, ** *p* < 0.01, *** *p* < 0.001 by ANOVA followed by Neuman Keuls test.

**Figure 3 biomolecules-14-00631-f003:**
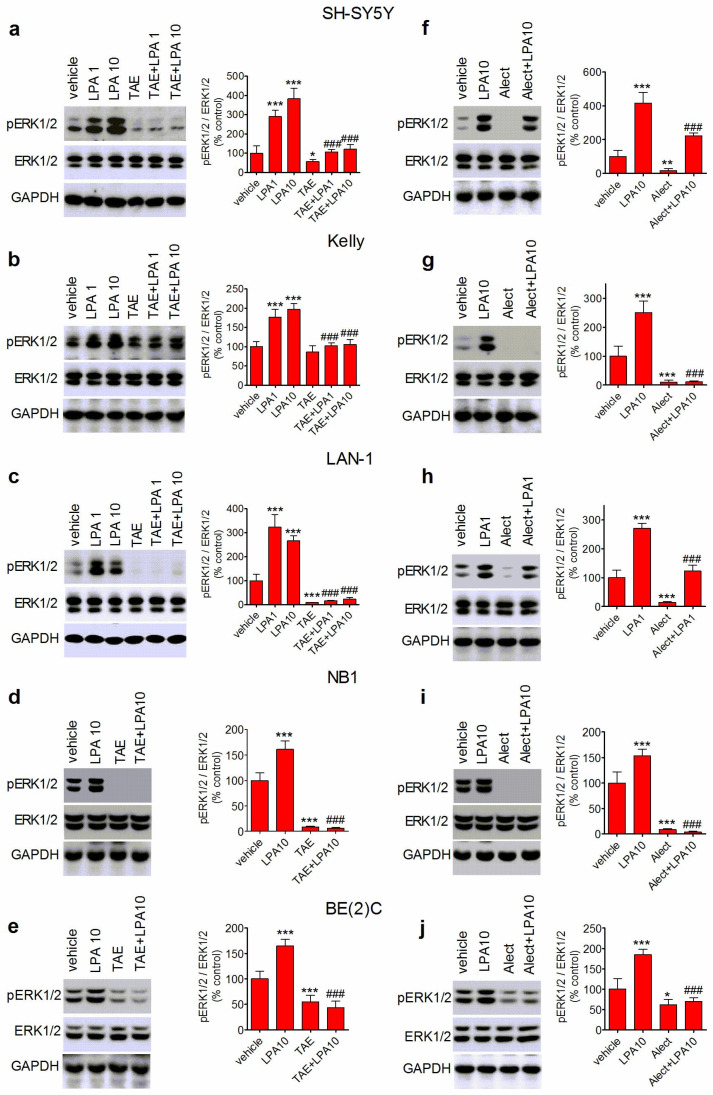
ALK inhibitors curtail LPA-induced ERK1/2 phosphorylation in human neuroblastoma cell lines. Cells were pretreated for 1 h with either vehicle or 50 nM TAE684 (TAE) (**a**–**e**), and for 6 h with either vehicle or 1 µM alectinib (alect) (**f**–**j**). Cells were then incubated for 5 min with either vehicle, LPA 1 µM (LPA 1), and/or LPA 10 µM (LPA 10). Cell lysates were analyzed for ERK1/2 phosphorylation by Western blot. Values are the mean ± SD of three independent experiments. * *p* < 0.05, ** *p* < 0.01, *** *p* < 0.001 vs. control (vehicle + vehicle); ^###^ *p* < 0.001 vs. vehicle + LPA by ANOVA followed by Neuman Keuls test.

**Figure 4 biomolecules-14-00631-f004:**
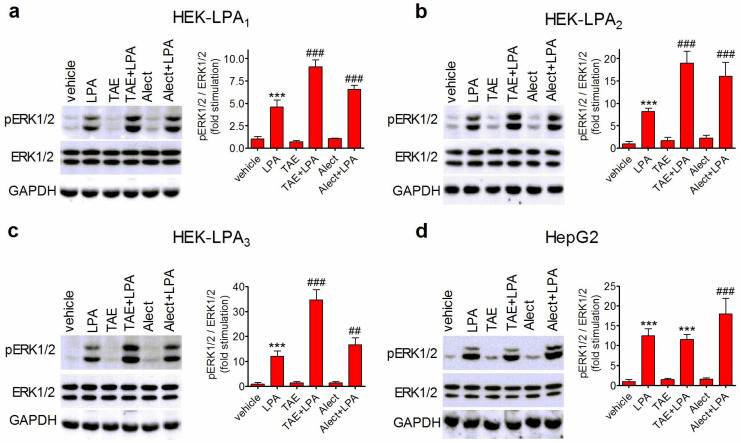
Effects of ALK inhibitors on LPA-induced ERK1/2 phosphorylation in HEK-LPA_1–3_ and HepG2 cells. Serum-starved HEK-293 cells stably transfected with *LPAR1* (HEK-LPA_1_), *LPAR2* (HEK-LPA_2_), and *LPAR3* (HEK-LPA_3_) genes and HepG2 cells were pretreated with either vehicle, 50 nM TAE684 (TAE) for 1 h, or 1 µM alectinib for 6 h and then incubated for 5 min with either 1 µM (**a**–**c**) or 10 µM (**d**) LPA. Cell lysates were analyzed for ERK1/2 phosphorylation by Western blot. Values are the mean ± SD of three independent experiments. *** *p* < 0.001 vs. control (vehicle + vehicle); ^##^ *p* < 0.01, ^###^ *p* < 0.001 vs. vehicle + LPA by ANOVA followed by Neuman Keuls test.

**Figure 5 biomolecules-14-00631-f005:**
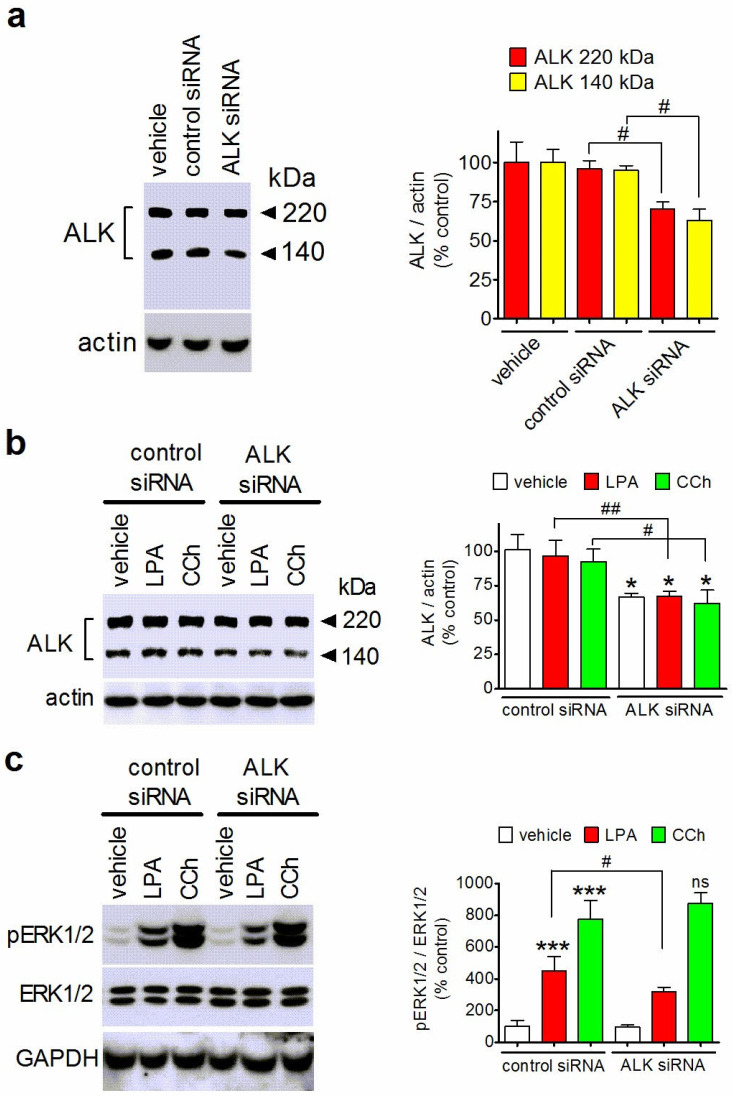
ALK depletion by siRNA treatment impairs LPA-induced ERK1/2 phosphorylation in SH-SY5Y cells. (**a**) Cells were treated with either transfectant alone (vehicle), control siRNA or ALK siRNA. Forty-eight hours after treatment, cell lysates were analyzed for ALK and actin levels by Western blot. Densitometric ratios are reported as percent of control (transfectant alone) and are the mean ± SD of three independent experiments. ^#^ *p* < 0.05 by ANOVA followed by Neuman Keuls test. (**b**,**c**) Cells were transfected with either control siRNA or ALK siRNA and then incubated for 5 min with either vehicle, 10 µM LPA or 10 µM CCh. Cell lysates were analyzed for ALK levels (**b**), and ERK1/2 phosphorylation (**c**). Values are expressed are percent of control (control siRNA + vehicle) and are the mean ± SD of four independent experiments. * *p* < 0.05, *** *p* < 0.001 vs. control. ^#^ *p* < 0.05, ^##^ *p* < 0.01; ns, not significantly different from control siRNA + LPA by ANOVA followed by Neuman Keuls test.

**Figure 6 biomolecules-14-00631-f006:**
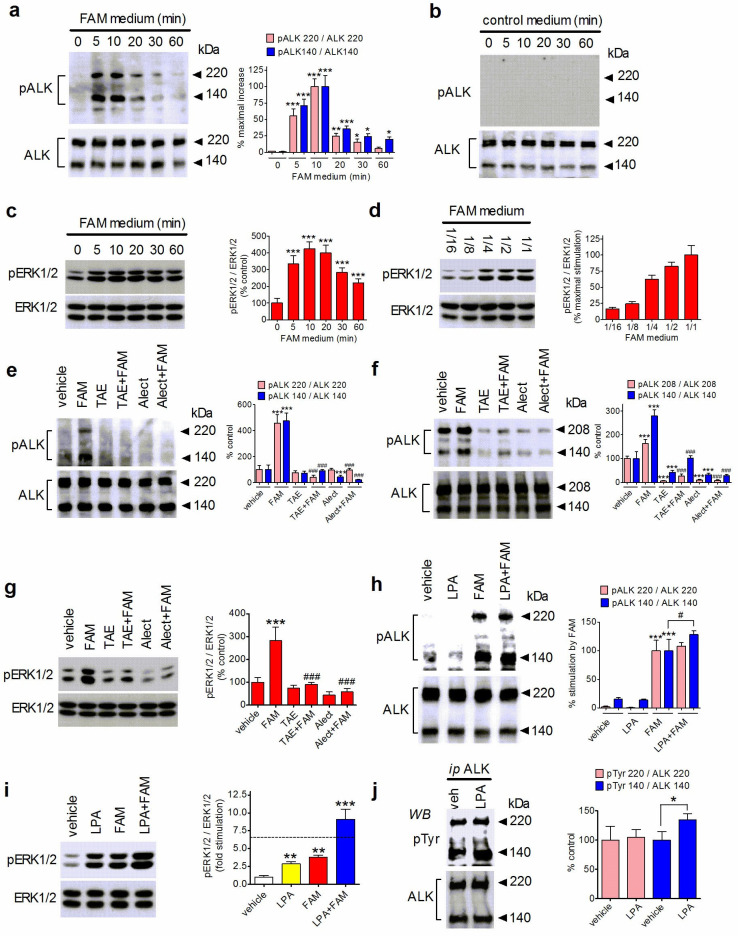
LPA potentiates ALK activation by FAM150B containing conditioned medium. (**a**,**b**) Serum-starved SH-SY5Y cells were incubated for the indicated periods of time with either FAM150B containing conditioned medium (FAM medium) (**a**) or medium of non-transfected HEK-293 cells (control medium) (**b**). Zero time samples were incubated with either control medium (**a**) or serum-free nonconditioned HEK-293 medium (**b**). Cell lysates were analyzed for phospho-ALK (Tyr1604) (pALK) and total ALK levels by Western blot. Densitometric ratios are reported percent of maximal ALK phosphorylation and are the mean ± SD of three independent experiments. * *p* < 0.05, ** *p* < 0.01, *** *p* < 0.001 vs. control (zero time samples) by ANOVA followed by Neuman Keuls test. (**c**) SH-SY5Y cells were incubated as indicated in (**a**) and cell lysates were analyzed for phospho-ERK1/2 and ERK1/2 levels. Values are the mean ± SD of three independent experiments. *** *p* < 0.001 vs. control (zero time) by ANOVA followed by Neuman Keuls test. (**d**) SH-SY5Y cells were incubated for 10 min with the indicated dilutions of FAM medium. Values are the mean ± SD of three experiments. (**e**,**f**) SH-SY5Y (**e**) and NB1 (**f**) cells were pretreated with either vehicle, 50 nM TAE684 (TAE) for 1 h, or 1 µM alectinib for 6 h, and then incubated in the presence of either control medium or FAM medium (FAM) for 10 min. Cell lysates were analyzed for phospho-ALK and total ALK. (**g**) SH-SY5Y cells were treated as indicated in (**e**) and cell lysates were analyzed for ERK1/2 phosphorylation. Values are the mean ± SD of three independent experiments. *** *p* < 0.001 vs. control (vehicle + control medium); ^###^ *p* < 0.001 vs. vehicle + FAM medium. (**h**,**i**) SH-SY5Y cells were pretreated for 5 min with either vehicle or 10 µM LPA and then incubated for 10 min with either control medium or FAM medium. Cell lysates were analyzed for phospho-ALK (**h**) and phospho-ERK1/2 (**i**). The dotted line in (**i**) indicates the level of pERK1/2 calculated by summing up the net stimulations by LPA and FAM medium alone. ** *p* < 0.01, *** *p* < 0.001 vs. control (vehicle + control medium), ^#^ *p* < 0.05 by ANOVA followed by Neuman Keuls test. (**j**) Serum- starved SY-SY5Y cells were treated for 5 min with either vehicle or 10 µM LPA. Cell lysates were subjected to immunoprecipitation with either anti-ALK antibody or preimmune antibody. Immunoprecipitates were analyzed for phospho-Tyrosine (pTyr) and ALK levels. No immunoreactive bands were detected in samples immunoprecipitated with preimmune antibody. Values are the mean ± SD of four experiments. * *p* < 0.05 by Student’s *t* test.

**Figure 7 biomolecules-14-00631-f007:**
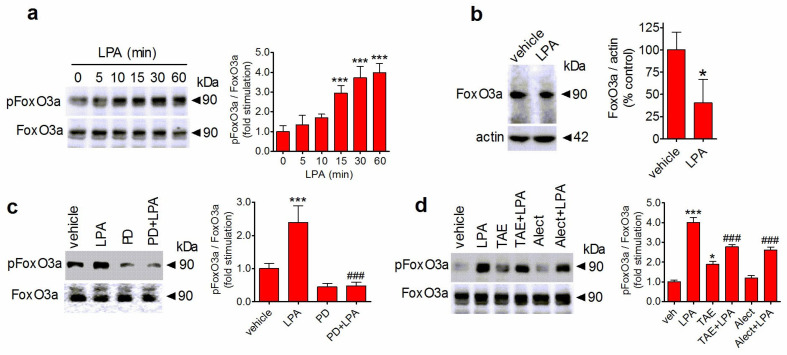
LPA triggers FoxO3a phosphorylation through ERK1/2 and ALK. (**a**) Serum-starved SH-SY5Y cells were incubated for the indicated periods of time with 1 µM LPA. Zero time samples were treated with vehicle and used as control. Thereafter, cell lysates were analyzed for phospho-FoxO3a (pFoxO3a) and total FoxO3a by Western blot. Values are the mean ± SD of three independent experiments. *** *p* < 0.001 by ANOVA followed by Neuman Keuls test. (**b**) SH-SY5Y cells were incubated for 24 h with either vehicle or 1 µM LPA and cell lysates were analyzed for FoxO3a levels. Values are the mean ± SD of three independent experiments. * *p* < 0.05 by Student’s *t* test. (**c**) SH-SY5Y cells were pretreated fo 1 h with either vehicle or 25 µM PD98059 (PD) and then incubated for 30 min with either vehicle or 1 µM LPA. (**d**) SH-SY5Y cells were pretreated with either vehicle, 50 nM TAE684 (TAE) for 1 h, or 1 µM alectinib for 6 h and then incubated for 30 min with either vehicle or 1 µM LPA. Values are the mean ± SD of three independent experiments. * *p* < 0.05, *** *p* < 0.001 vs. control (vehicle + vehicle). ^###^ *p* < 0.001 vs. vehicle + LPA by ANOVA followed by Neuman Keuls test.

## Data Availability

The data presented in this study are available on request from the corresponding author.

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
