# Peer review of "Lysophosphatidic Acid Stimulates Mitogenic Activity and Signaling in Human Neuroblastoma Cells through a Crosstalk with Anaplastic Lymphoma Kinase"

_biomolecules, 2024, doi:10.3390/biom14060631_

Round 1
Reviewer 1 Report
Comments and Suggestions for Authors
This manuscript presents a crosstalk between LPA and ALK, which induces mitogenesis in neuroblastoma. Overall, this is a well-designed mechanistic study. However, there are some concerns that need to be addressed before publication.
Major Points:
1. The authors' data indicate that ALK is involved in the cell proliferative effects of LPA, but there is a lack of knowledge about the underlying mechanism between LPA and ALK, as stated in the Discussion section. Even if the details are to be worked out in the future, it would be necessary at the very least to verify whether LPA receptors (GPCR signaling) are involved. In this case, is it due to Gi signaling, as was the case with IGF-IR? Also, would the responsible receptor be LPA2? Or are there other transactivation mechanisms at play?
2. In Figure 2, each of the five cells shows different sensitivity to the phosphorylation of ERK by LPA; differences in ALK genomic status do not seem to explain this difference. Do you have any comments on what might account for this difference? Is it due to differential expression levels of the LPA receptor?
3. In Figure 3c (LAN-1 cell), although there does not appear to be a significant difference in pERK levels between the control (vehicle) and LPA-treated groups, at least in the Western blot bands, quantitative results show a 3-fold increase in the LPA-treated group. Please confirm that this quantitative result is indeed correct.
4. In Figure 5, the ALK knockdown efficiency (reduced by about 30%) is very low. As a result, the effect is minor, although the authors claim it to be a significant difference (Figure 5c). Thus, it seems difficult to conclude whether the action of LPA is canceled by ALK knockdown.
Minor Points:
- On page 11, line 318, please correct the notation of LPA receptor genes (LPA1R -> LPAR1, LPA2R -> LPAR2, LPA3R -> LPAR3).
Reviewer 2 Report
Comments and Suggestions for Authors
Comments: (this manuscript has many errors/flaws)
1. Table 5: the siRNA knockdown is not working based on the results. The density of bands on the western blots look identical, no change. Please explain. Without this result, the conclusion of the current manuscript is weak.
2. The authors need to show the shorter exposure of the p-ERK WB bands. Some of the bands look similar in density, no change.
3. The labels on Figure 3C are not correct.
4. Authors need to have abbreviation section to allow readers to understand abbreviations easily.
5. Authors used different doses of LPA and drugs in different experiments. Need explanation.
6. Please show siRNA sequences.
7. Does serum starvation only affect p-ERK levels in tested cell lines? This will be an important control.
8. On Figure 1B: TAE+LAP group seems to have fewer colonies? Please explain.
9. On Figure2, 3, 4, and 5: Please add beta actin or GAPDH as loading control.
Comments on the Quality of English LanguageMinor editing of English language required
Round 2
Reviewer 1 Report
Comments and Suggestions for Authors
None
Author Response
The Reviewer made no additional comments. Therefore, we have no answer to provide.
Reviewer 2 Report
Comments and Suggestions for Authors
The main issue of the study is low reduction by siRNA experiments. only 25% reduction as quantification of bands shows. Anyway to improve this? using new siRNA or create stable cell line?
Author Response
In the second review, the Referee argued that the siRNA treatment only caused a 25 % reduction in ALK levels “as quantification of bands shows”. However, as pointed out in the reply to the first review (point 1) and reported in the manuscript, in the experiments shown in Figure 5a densitometric ratios of ALK/actin indicated that ALK siRNA treatment reduced the levels of the 220 and 140 kDa forms of ALK by 29 ± 5 and 37 ± 8 %, respectively. In the experiments reported in Figure 5b, cells treated with ALK siRNA showed a decrease of total ALK levels by an average of 34 %. These changes were statistically significant and associated with a significant reduction in the stimulation of phospho-ERK1/2 by LPA. We therefore think that these results are sufficient to provide additional evidence that LPA signaling is dependent on ALK. Most importantly, the partial reduction in ALK levels obtained by the siRNA treatment was not accompanied by changes in cell viability, as also indicated by the finding that the response to CCh was not altered. As mentioned in the first response to the Referees’ comments, a number of independent studies have shown that in neuroblastoma cells a complete or near-complete ALK genetic depletion causes growth inhibition and loss of viability (Refs. 15, 17,18, 33, 34, 58). In particular, it has been reported that in SH-SY5Y cells ALK knockdown causes apoptotic death (Refs 18, 58).
We agree with the Referee that we may get a greater reduction of ALK expression by using new siRNA or developing a stable cell line lacking ALK, but this would be likely accompanied by an impairment of cell viability, a condition that does not allow the study of LPA-ALK crosstalk.